# The Positive Association between Muscle Mass and Bone Status Is Conserved in Men with Diabetes: A Retrospective Cross-Sectional and Longitudinal Study

**DOI:** 10.3390/jcm11185370

**Published:** 2022-09-13

**Authors:** Hyun Uk Moon, Seung Jin Han, Hae Jin Kim, Yoon-Sok Chung, Dae Jung Kim, Yong Jun Choi

**Affiliations:** Department of Endocrinology and Metabolism, Ajou University School of Medicine, Suwon 16499, Korea

**Keywords:** muscle mass, bone, appendicular skeletal muscle index, bone mineral density, trabecular bone score

## Abstract

Bone and muscle are known to be correlated and interact chemically each other. Diabetes affects the health status of these two types of organ. There has been lack of studies of men on this topic. This study aims to investigate the relationship between bone and muscle status in men with and without diabetes. This study enrolled 318 and 88 men with and without diabetes, respectively, between April 2007 and December 2017. The appendicular skeletal muscle index (ASMI) was correlated with femoral neck bone mineral density (BMD), total hip BMD, and the trabecular bone score (TBS) in both groups (*p* < 0.001–0.008). In analysis of the changes in muscle mass and bone-related parameters over the 3 years, the ASMI was correlated with total hip BMD only in diabetes group (*p* = 0.016) and the TBS in both groups (*p* < 0.001–0.046). This study showed that the positive correlation between muscle mass and bone status was largely conserved in diabetic group in men. Moreover, in a long-term perspective, muscle mass might be more correlated with the bone microarchitecture or bone quality than bone density, and the association between muscle mass and total hip BMD could be stronger in the diabetic group.

## 1. Introduction

Due to the increase in life expectancy, osteoporosis and sarcopenia have become a worldwide public health problem [1,2]. Osteoporotic fractures increase the risk of morbidity, mortality, and medical costs [3], which is reflected in the higher socioeconomic burden of fragility fractures and incidence rates in Asia [4]. Sarcopenia, which is characterized by a lack of muscular mass, strength, and function, ultimately results in frailty and may result in worse outcomes (e.g., fracture, disability, hospitalization, and death) [5,6]. Recent studies have reported biochemical interactions between the skeletal muscle and bone [7,8]. Through mechanical loading, skeletal muscles locally communicate with the bone as a paracrine organ [9], while muscular tissue acts as an endocrine organ that affects bone metabolism through circulation, which is defined as systemic interaction [10,11].

Diabetes mellitus (DM) is a chronic disease which can cause bone fragility and muscle loss. Multiple studies have shown that high glucose level, and the metabolic products of diabetes are risk factors of fragility fractures [12,13], while insulin resistance or oxidative stress result in muscle loss [14]. Therefore, not only the unique interaction between muscle and bone but also DM affects the bone status and muscle mass. However, it has not yet been clearly shown whether the systemic or local interaction between muscle and bone is conserved in patients with DM. 

Various clinical studies have focused on bone health, especially in postmenopausal women. Considering the substantial metabolic changes after menopause, women are more vulnerable to osteoporosis than men [15]. Although they are comparatively less influenced by such dramatic hormonal changes and less at risk for bone fragility, men are more susceptible to muscle loss than women [16]. However, there have been few studies on the association between muscle and bone in men. To our knowledge, no specific study has compared the interaction of muscle and bone between men with and without diabetes. Therefore, we investigated the association between muscle mass and bone-related parameters such as trabecular bone score (TBS) and bone mineral density (BMD) in male patients with DM and compared the clinical data with that of non-DM group. In addition, we examined the effect of changes in muscle mass over time on the TBS and BMD based on follow-up data.

## 2. Materials and Methods

### 2.1. Study Design and Population

In this retrospective cross-sectional study, we screened all male patients with type-2 DM who visited the endocrine clinic and underwent a BMD measurement at Ajou University Hospital between April 2007 and December 2017. In the present study, type-2 DM was defined as diabetes that began after the age of 30 years and was supported by medical records that matched the features of type-2 DM. In this research, we used the 2022 American Diabetes Association (ADA) definition for diabetes as follows: fasting plasma glucose ≥ 126 mg/dL, glycated hemoglobin (HbA1C) ≥ 6.5%, or 2-h plasma glucose ≥ 200 mg/dL during oral glucose tolerance test, or in a patient with classic symptoms of hyperglycemia or hyperglycemic crisis, a random plasma glucose ≥200 mg/dL [17]. The mean duration of DM was 7.87 years. All patients enrolled in this study underwent at least one follow-up BMD test, with simultaneous measurement of TBS and body composition, within 3 years of the initial BMD test and after at least a 1-year interval. The mean follow-up period of BMD measurement in DM and non-DM groups were 1.94 years and 1.63 years, respectively.

We excluded patients who had taken drugs for osteoporosis or diabetes that could affect bone or muscle metabolism (e.g., a bisphosphonate, selective estrogen receptor modulator, and thiazolidinedione). Patients having missing data on BMD, TBS, body composition data, HbA1C level, lipid profile, or serum creatinine level were also excluded. Finally, 318 men with diabetes were selected for this study. Simultaneously, we additionally collected data of 88 men without diabetes who matched the aforementioned criteria. Men in this group underwent a BMD test for a periodic medical checkup, and patients with hyperthyroidism, hyperparathyroidism, chronic kidney disease, rheumatoid arthritis, and Cushing disease were excluded (Figure 1). This study was approved by the ethical review board of this institution (AJIRB-MED-MDB-21-318) and followed the Declaration of Helsinki.

### 2.2. Data Collection

We collected baseline laboratory data of the patients including the HbA1c level, serum creatinine level, and lipid profile (total cholesterol, high-density lipoprotein cholesterol [HDL-C], and triglyceride [TG] levels). We also recorded the prescription history of anti-diabetic agents and osteoporosis medications from the first BMD test date to the next follow-up BMD test date. 

### 2.3. Measurements

Each participant’s physical profile, which included height, weight, and body mass index (BMI) was assessed using standard protocols. The lipid profile, and the level of serum creatinine and HbA1c were measured using automated procedures at the laboratory of Ajou University Hospital. The BMDs (g/cm^2^) of the lumbar spine from L1 through L4 (L1–L4), femoral neck, and total hip were evaluated by dual energy X-ray absorptiometry (DXA) (Lunar Prodigy; GE Healthcare, Madison, WI, USA) and analyzed with the enCORE Software Platform version 16.0 (General Electric Medical Systems, Madison, WI, USA). Affected vertebrae, such as those having compression fractures or degenerative alterations, were excluded from the calculation of BMD in each bone. The precision errors (%CV = standard deviation/mean × 100) for the lumbar spine BMD and femoral neck BMD were 0.87% and 0.93%, respectively. The least significant change (LSC) was 0.024 g/cm^2^ for the lumbar spine BMD and 0.026 g/cm^2^ for the femoral neck BMD. TBS iNsight version 3.0.2.0 (Med-Imaps, Plan-les-Ouates, Switzerland) was used to evaluate the TBS retrospectively. The program used the spine raw DXA images for the same regions of measurement as those used to estimate the lumbar spine BMD. The lumbar spine TBS was computed as the mean value of individual measurements for L1–L4. For TBS, the precision error and LSC were 1.408% and 0.039, respectively.

Body composition was also assessed using DXA and analyzed using encore Software Platform version 16.0 (General Electric Medical Systems). Lean mass was measured and separated into trunk, android, gynoid and appendicular components. The appendicular skeletal muscle index (ASMI) as an indicator of sarcopenia was calculated using the following formula: ASMI = (appendicular skeletal muscle mass [kg]/square of height [m^2^]) [18]. Appendicular skeletal muscle mass was calculated as the sum of the lean soft tissue mass in the arms and legs. All measurements of BMD, TBS, and body composition data were performed according to the manufacturer’s guidelines, by a highly trained technician.

### 2.4. Statistical Analysis

All data of clinical characteristics are expressed as median values with interquartile range or frequency (percentage). Mann-Whitney U test and chi-squared test were used to compare differences between the DM and non-DM groups. Pearson correlation analysis was used to determine the associations between muscle-related parameter (ASMI), bone-related parameters (BMD and TBS), and clinical characteristics. Multiple linear regression analyses were performed using the initial BMD or TBS as the dependent variable and ASMI as independent variable with adjustment for age, BMI, and levels of serum creatinine, TG, HDL-C, and HbA1c. For the next step, we tried to evaluate the correlation between the change in muscle mass and bone-related parameters using the longitudinal follow-up data. A generalized estimating equation (GEE) analysis was performed on the changes of TBS, BMD, and ASMI after adjustment for age; BMI; and levels of serum creatinine, TG, HDL-C, and HbA1c. We compared the initial measurement data and follow-up data of body profiles, ASMI, and bone-related parameters using Wilcoxon signed-rank test. All statistical analyses were performed using SPSS version 25.0 (IBM Corp., Armonk, NY, USA). Statistical significance was set at *p* < 0.05.

## 3. Results

### 3.1. Basal Characteristics of DM and Non-DM Groups

The characteristics of the enrolled patients are presented in Table 1. In total, 406 men (DM group, 318 and non-DM group, 88) were included in this study. The age of participants in the DM group was higher than that of participants in the non-DM group (52.00 vs. 46.50 years, *p* < 0.001). The mean weight and BMI were also higher in the DM group than in the non-DM group (73.40 vs. 70.00 kg, *p* = 0.002, and 25.45 vs. 24.00 kg/m^2^, *p* < 0.001, respectively). The ASMI was higher in the DM group than in the non-DM group (7.775 vs. 7.341 kg/m^2^, *p* = 0.002). The DM group also had higher lumbar spine and femur-related BMDs than the non-DM group (*p* < 0.001), but the TBS was not different between the groups (*p* = 0.454). The proportion of patients with osteoporosis was higher in the non-DM group than in the DM group (15.9% vs. 2.8%, *p* < 0.001). The non-DM group had higher total cholesterol and HDL-C levels (*p* < 0.001–0.013) but lower TG levels than the DM group (*p* < 0.001). There was no significant difference in mean creatinine levels between the groups (*p* = 0.950). 

### 3.2. Relationship between Muscle and Bone-Related Parameters Based on Initial Measurement Data 

A simple correlation analysis was performed to evaluate the association between muscle and bone-related parameters (Table 2). Based on initial measurement data, the lumbar spine BMD, femoral neck BMD, total hip BMD, and TBS were correlated with the ASMI (γ = 0.163–0.452, *p* < 0.001–0.004) in the DM and non-DM groups. Among the metabolic parameters, BMI showed a significant relationship with ASMI (γ = 0.778–0.835, *p* < 0.001) in both groups.

We further performed multiple regression analysis for the muscle mass and bone-related parameters with adjustment for age, BMI, levels of creatinine, HDL-C, TG, and HbA1c. Table 3 shows the association between ASMI and bone-related parameters. Model 1 was adjusted for age and levels of creatinine, TG, HDL-C, and HbA1c. Model 2 was adjusted for BMI in addition to model 1 adjustments. Regarding the relationship between ASMI and bone-related parameters, in model 1, all BMD parameters and TBS showed significant positive relationships with ASMI (DM group: lumbar spine BMD, β = 0.174, *p* = 0.003; femoral neck BMD, β = 0.331, *p* < 0.001; total hip BMD, β = 0.405, *p* < 0.001; TBS, β = 0.203, *p* < 0.001 and non-DM group: lumbar spine BMD, β = 0.302, *p* = 0.006; femoral neck BMD, β = 0.409, *p* < 0.001; total hip BMD, β = 0.428, *p* < 0.001; TBS, β = 0.357, *p* = 0.002). Model 2 showed results similar to those of model 1 concerning the association between ASMI and bone-related parameters, except that lumbar spine BMD was not correlated with ASMI in DM group (DM group: lumbar spine BMD, β = 0.187, *p* = 0.076; femoral neck BMD, β = 0.389, *p* < 0.001; total hip BMD, β = 0.487, *p* < 0.001; TBS, β = 0.313, *p* = 0.002 and non-DM group: lumbar spine BMD, β = 0.481, *p* = 0.005; femoral neck BMD, β = 0.622, *p* < 0.001; total hip BMD, β = 0.663, *p* < 0.001; TBS, β = 0.464, *p* = 0.008).

### 3.3. Relationship between the Changes in Muscle Mass and Bone-Related Parameters

Subsequently, we performed a GEE analysis of the changes in BMDs (ΔBMDs), TBS (ΔTBS), and ASMI (ΔASMI) which were estimated within the 3-year follow-up period. Appendix A shows the time-dependent change of BMDs, TBS, and ASMI. Table 4 presents the results of the GEE analysis. ΔASMI was correlated with the Δtotal hip BMD in DM group, but not in non-DM group in model 1 (DM group: B = 0.012, *p* = 0.006 and non-DM group: B = 0.015, *p* = 0.157) and model 2 (DM group: B = 0.013, *p* = 0.016 and non-DM group: B = 0.009, *p* = 0.065). ΔASMI was significantly associated with ΔTBS in DM and non-DM groups in both model 1 (DM group: B = 0.011 *p* = 0.040 and non-DM group: B = 0.019, *p* < 0.001) and model 2 (DM group: B = 0.024, *p* = 0.046 and non-DM group: B = 0.020, *p* < 0.001). 

## 4. Discussion

### 4.1. Summary of Findings

In men, ASMI was significantly correlated with most of the bone-related parameters including TBS, except for the lumbar spine BMD in DM group. The analysis of changes in muscle mass and bone-related parameters showed that ΔASMI was associated with ΔTBS in both DM and non-DM groups and had a significant relationship with Δtotal hip BMD only in the DM group. To our best knowledge, this is the first report on the association between bone and muscle mass in men with and without DM. 

### 4.2. Muscle Mass and Bone-Related Parameters in Male Group

Based on the multiple regression analysis using initial measurement data, higher ASMI was associated with higher femoral neck BMD, total hip BMD, and TBS in both DM and non-DM groups. ASMI had a significant relationship with lumbar spine BMD in non-DM group. On the other hand, a study done by Kwak and colleagues has shown that muscle mass is positively associated with femoral neck BMD or total hip BMD but not with the lumbar spine BMD or TBS in postmenopausal women [19]. However, our results differ from those of the previous study in that the lumbar spine BMD in non-DM group and TBS in both groups were correlated with muscle mass. The difference in study design, muscle measurement devices or statistical adjusting factors could be a reason for this inconsistent result. However, physical differences in gender could offer a logical explanation. First, females have a more decreased vertebral trabecular and cortical bone mass compared to males. Men had less age-related vertebral bone loss, particularly in the cortical bone [20]. More proportion and less degradation of the cortical bone in spine might maintain more local interaction with muscle in men. Second, the lesser degree of spine degeneration in men than in women could be another reason for the different results [21]. Finally, the lack of a sudden change in gonadal hormone levels in men might be another reason for this outcome. A decrease in serum estrogen levels in women is a massive systemic endocrine change that disturbs the interaction between the lumbar spine and muscle mass [22,23,24]. Moreover, one report showed that skeletal muscle was correlated with lumbar spine BMD in premenopausal women, which is in concert with our study findings [25]. Additional research is needed to provide evidence of the effect of gonadal hormones such as estrogen or testosterone on the relationship between muscle and bone in male with and without DM.

### 4.3. Differences between the DM and Non-DM Groups

ASMI was significantly correlated with most of the bone-related parameters including TBS in DM group, except for the lumbar spine BMD. Diabetes could be responsible for this result. There are numerous reports on the effects of DM on bone and muscle. Diabetes is a known risk factor of fractures or sarcopenia [12,26]. Osmotic stress due to hyperglycemia and advanced glycosylation-end products are factors that induce bone fragility [12,27,28]. In contrast, insulin resistance can decrease protein synthesis and increase protein degradation, which can induce muscle loss [14,29,30]. Reduced insulin signaling and increased levels of inflammatory cytokines such as interleukin (IL)-1, IL-6, and tumor necrosis factor-alpha have been suggested to be associated with sarcopenia [31,32]. Chronic inflammation, oxidative stress, and mitochondrial dysfunction have also been suggested as mechanisms of sarcopenia in diabetes [33,34]. Recent studies have indicated that undercarboxylated osteocalcin has a major role in increasing the muscle glucose uptake and muscle mass [35]. Exercise and ageing are relevant to the concentration of circulating undercarboxylated osteocalcin [36,37]. These systemic changes in patients with DM might affect the relationship between skeletal muscle mass and bone microarchitecture. 

Before adjusting for BMI, ASMI was correlated with the lumbar spine BMD in DM group. However, after adjustment for BMI, ASMI did not show a significant relationship with the lumbar spine BMD, while lumbar spine BMD in non-DM group and all the other bone-related parameters maintained a significant correlation with ASMI. There may be possible causes for the different results between DM and non-DM groups. First, the ASMI may not accurately indicate sarcopenia in obese or overweight individuals [38], and greater body fat represented by a higher BMI, exerts adverse effects on bone and skeletal muscle [39,40]. These factors could also be applied to the DM group that had more obese population (median BMI, DM vs. non-DM, 25.45 kg/m^2^ vs. 24.00 kg/m^2^, *p* < 0.001, Table 1). Second, approximately 70% of men in the DM group did not show muscle mass loss in the sarcopenia range, and the mean value of the ASMI in the DM group was even higher than that in the non-DM group in this study, which is inconsistent with findings of previous study showing that diabetes increases the risk of muscle loss and frailty [26]. Continuous management which includes not only medical treatment but also lifestyle modification will result in preserving or increasing muscle mass in DM patients. However, further studies with more subjects and more accurate muscle data are required to find the factors responsible for those inconsistent results. 

Our study showed that lumbar spine BMD and femur-related BMDs were higher in the DM group than in the non-DM group (Table 1). Although a high glucose level is a risk factor of bone fragility, lumbar spine BMD along with other BMD parameters do not always accurately represent bone health in the clinical setting of DM [13,41,42,43]. An increase in stress and mechanical loading on the bone can be considered as a factor for increased BMD in DM patients [44,45].

### 4.4. Association between the Changes in Muscle and Bone Parameters

We additionally analyzed the association between changes in ASMI, and bone parameters using follow-up data. ΔASMI showed a significant relationship with ΔTBS in the DM and non-DM groups, which is similar to the results of our former analysis using the initial measurement data. In contrast, no correlation was shown between ΔASMI and Δlumbar spine BMD in both groups. TBS is known to be more related to the microarchitecture of bone than lumbar spine BMD [46]. It could be suggested that muscle mass might affect the bone microarchitecture or bone quality more than bone density in the view of long-term management. However, to confirm this theory, further studies with more participants and more longitudinal follow-up data are needed.

ΔASMI was correlated with Δtotal hip BMD in the DM group, but no significant correlation was found in the non-DM group. After adjustment for BMI, statistical significance was increased, but not sufficiently to make a meaningful correlation between ΔASMI and Δtotal hip BMD in the non-DM group (*p* = 0.065–0.157). On the other hand, whether adjusting the BMI or not, ΔASMI was not associated with Δfemur neck BMD in both groups. The reason for the different results between the analysis of the initial measurement data and that of the changes is unclear, but there have been multiple reports on the limits of the femoral neck BMD. The precision of measuring bone density at the femoral neck site is relatively poor [47], and the femoral neck subregion is smaller and more variable than the total hip measurement; therefore, its reproducibility is not sufficient to minimize measurement error [48,49]. In addition, the different proportions of cortical bone between femoral neck and total hip bone could be suggested as another reason for the different results [50]. 

### 4.5. Strengths and Limitations

This study has several strengths. First, this research focused on men, and we collected data from a sufficient study population. Considering the higher prevalence of sarcopenia in men than in women [16], further prospective and clinical trials on muscle-bone-related research targeting men are warranted. Second, this study enrolled patients with diabetes who were vulnerable to osteoporosis and sarcopenia which represents the clinical setting with much accuracy. Third, we used not only the one-time data but also the follow-up data of each man. This made it possible to check whether the relationship between bone and muscle mass was maintained in the situation of long-term management. 

Despite these strengths, this study has several limitations. First, this research was based on a cross-sectional study of a Korean tertiary hospital-based cohort; hence, it could not establish a causal association. Second, some heterogeneity existed in the follow-up period, wherein a broad interval of 1–3 years was set. The similarity and stability of blood glucose level or blood glucose level changes might not be strictly ensured. This point could cause potential errors in our data analysis. However, all diabetic patients in our study group were treated in the endocrinology department of a tertiary hospital according to the standard protocols and guidelines. The mean duration of DM was not short, over seven years, and HbA1C level was measured every 3 to 6 months (mean HbA1C = 7.6%). In addition, there was no first-diagnosed diabetes patient in the study group. Therefore, the fluctuation range of glucose level might not lead to significant errors in the current analysis. Third, due to the DM vs. non-DM study setting in one specific hospital, selection bias or Berksonian bias could occur. Fourth, other biomarkers such as the vitamin D and calcium levels, or bone turnover markers were not available in our study population. In addition, as there were no data of gonadal hormones such as estrogen or testosterone level, the role of gonadal hormones in the relationship between muscle and bone could not be determined. Finally, there is still uncertainty in the accuracy of the muscle mass measurement. Computed tomography or magnetic resonance imaging is the gold standard for quantifying muscle mass. However, DXA is an accurate tool for measuring body composition compared to bioelectrical impedance analysis, and it has advantages such as cost-effectiveness, easy applicability, and low radiation [51].

## 5. Conclusions

In this study, muscle mass showed significant relationship with the bone status in men with diabetes. Positive association between ASMI and bone-related parameters such as TBS, femoral neck BMD, and total hip BMD was observed in DM group, which was similar to that in non-DM group. This result suggests that systemic or local interaction between muscle and bone is conserved in male patients with diabetes. Regarding time-dependent changes, the only bone-related parameter correlated with ASMI in both DM and non-DM groups was TBS. It seems that muscle mass might be more associated with the bone microarchitecture or bone quality. In addition, ASMI maintained significant relationship with total hip BMD in DM group. Further clinical studies based on these findings will help healthcare providers to manage diabetic patients who are prone to osteoporosis or sarcopenia. 

## Figures and Tables

**Figure 1 jcm-11-05370-f001:**
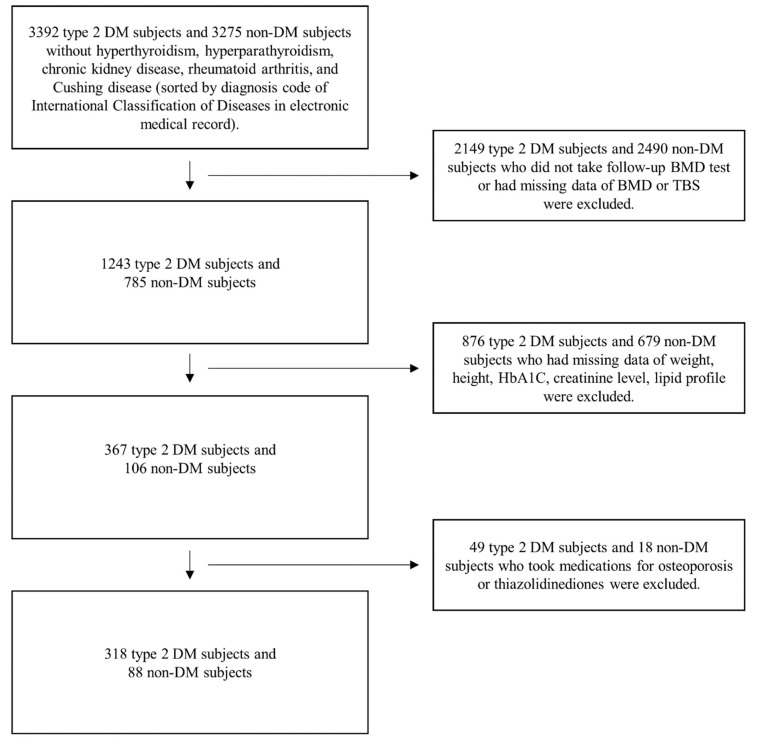
Flow diagram of participant inclusion and exclusion criteria. DM: diabetes mellitus, BMD: bone mineral density, TBS: trabecular bone score.

**Table 1 jcm-11-05370-t001:** Descriptive characteristics of men with diabetes or non-diabetes included in the study.

Characteristic	DM (*n* = 318)	Non-DM (*n* = 88)	*p* Value
Age (years)	52.000 (45.750–59.250)	46.500 (36.000–53.000)	<0.001
Height (cm)	169.600 (166.100–173.125)	172.250 (167.550–176.275)	0.002
Weight (kg)	73.400 (66.075–82.025)	70.000 (61.725–76.975)	0.002
BMI (kg/m^2^)	25.450 (23.700–28.025)	24.000 (20.650–25.275)	<0.001
ASMI (kg/m^2^)	7.775 (7.133–8.333)	7.341 (6.752–8.098)	0.002
Lumbar spine BMD (g/cm^2^)	1.141 (1.024–1.268)	0.994 (0.905–1.182)	<0.001
Femoral neck BMD (g/cm^2^)	0.962 (0.864–1.059)	0.919 (0.829–1.028)	0.042
Total hip BMD (g/cm^2^)	1.041 (0.934–1.144)	0.951 (0.842–1.030)	<0.001
Lumbar spine T-score	−0.326 (−1.300–0.730)	−1.512 (−2.265–0.027)	<0.001
Femoral neck T-score	0.090 (−0.658–0.841)	−0.161 (−0.930–0.599)	0.050
Total hip T-score	0.766 (−0.051–1.565)	0.074 (−0.764–0.688)	<0.001
TBS	1.458 (1.396−1.517)	1.460 (1.407–1.520)	0.454
Creatinine (mg/dL)	0.990 (0.900–1.140)	1.000 (0.900–1.120)	0.950
TG (mg/dL)	137.500 (95.750–189.750)	98.500 (68.000–162.250)	<0.001
HDL-C (mg/dL)	42.000 (36.000–51.000)	51.000 (43.500–60.000)	<0.001
Total cholesterol (mg/dL)	161.000 (136.000–187.000)	170.000 (150.250–197.750)	0.013
HbA1C (%)	7.600 (7.000–9.125)	-	-
Osteoporosis, n (%)	9 (2.8%)	14 (15.9%)	<0.001

Data presented as median (IQR). DM: diabetes mellitus, BMI: body mass index, ASMI: appendicular skeletal muscle mass index, BMD: bone mineral density, TBS: trabecular bone score, TG: triglyceride, HDL-C: high density lipoprotein cholesterol.

**Table 2 jcm-11-05370-t002:** Simple correlation of ASMI with bone-related parameters and metabolic factors.

	ASMI of DM (*n* = 318)		ASMI of Non-DM (*n* = 88)	
Variables	γ	*p*	γ	*p*
Initial lumbar spine BMD	0.163	0.004	0.313	0.003
Initial femoral neck BMD	0.379	<0.001	0.425	<0.001
Initial total hip BMD	0.412	<0.001	0.452	<0.001
Initial TBS	0.238	<0.001	0.366	<0.001
Age	−0.256	<0.001	−0.030	0.779
BMI	0.835	<0.001	0.778	<0.001
HbA1C	−0.054	0.335	−0.013	0.903
Creatinine	0.003	0.956	0.248	0.020
TG	0.046	0.418	0.170	0.113
HDL-C	−0.075	0.183	−0.301	0.004

DM: diabetes mellitus, ASMI: appendicular skeletal muscle index, BMD: bone mineral density, TBS: trabecular bone score, BMI: body mass index, TG: triglyceride, HDL-C: high density lipoprotein cholesterol.

**Table 3 jcm-11-05370-t003:** Multiple regression analysis of bone-related parameters on ASMI.

		ASMI of DM (*n* = 318)			ASMI of Non-DM (*n* = 88)		
Variables		β	t	*p*	β	t	*p*
Initial lumbar spine BMD	Model 1	0.174	3.047	0.003	0.302	2.798	0.006
	Model 2	0.187	1.779	0.076	0.481	2.878	0.005
Initial femoral neck BMD	Model 1	0.331	6.185	<0.001	0.409	3.798	<0.001
	Model 2	0.389	3.963	<0.001	0.622	3.759	<0.001
Initial total hip BMD	Model 1	0.405	7.554	<0.001	0.428	4.103	<0.001
	Model 2	0.487	4.957	<0.001	0.663	4.157	<0.001
Initial TBS	Model 1	0.203	3.623	<0.001	0.357	3.235	0.002
	Model 2	0.313	3.052	0.002	0.464	2.698	0.008

Model 1: Adjusted for age, creatinine, TG, HDL-C, and HbA1C. Model 2: Adjusted for variables in model 1 as well as BMI. Independent variable: ASMI; Dependent variable: BMD parameters and TBS. DM: diabetes mellitus, ASMI: appendicular skeletal muscle index, BMD: bone mineral density, TBS: trabecular bone score. β: regression coefficient, t: coefficient divided by its standard error.

**Table 4 jcm-11-05370-t004:** GEE model of the association between the change in bone-related parameters and ASMI.

Variables		B		Std.err		Wald		*p*-Value	
Dependent	Independent (ΔASMI)	DM	Non-DM	DM	Non-DM	DM	Non-DM	DM	Non-DM
		(*n* = 318)	(*n* = 88)	(*n* = 318)	(*n* = 88)	(*n* = 318)	(*n* = 88)	(*n* = 318)	(*n* = 88)
ΔLumbar spine BMD	Model 1	0.011	0.015	0.007	0.014	2.950	1.055	0.086	0.304
	Model 2	0.009	0.006	0.010	0.006	0.919	1.014	0.338	0.314
ΔFemoral neck BMD	Model 1	0.006	0.012	0.005	0.011	1.081	1.213	0.298	0.271
	Model 2	0.005	0.004	0.007	0.004	0.390	1.314	0.532	0.252
ΔTotal hip BMD	Model 1	0.012	0.015	0.004	0.011	7.432	2.003	0.006	0.157
	Model 2	0.013	0.009	0.006	0.005	5.761	3.407	0.016	0.065
ΔTBS	Model 1	0.011	0.019	0.005	0.005	4.229	17.399	0.040	<0.001
	Model 2	0.024	0.020	0.012	0.005	3.980	14.431	0.046	<0.001

GEE: generalized estimating equations. Model 1: Adjusted for age, creatinine, TG, HDL-C, and HbA1C. Model 2: Adjusted for variables in model 1 as well as change in BMI between follow-up period. DM: diabetes mellitus, ASMI: appendicular skeletal muscle index, BMD: bone mineral density, TBS: trabecular bone score. B: standardized beta coefficient, Std.err: standard error.

## Data Availability

Not applicable.

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
