# Peer review of "The Positive Association between Muscle Mass and Bone Status Is Conserved in Men with Diabetes: A Retrospective Cross-Sectional and Longitudinal Study"

_jcm, 2022, doi:10.3390/jcm11185370_

Round 1
Reviewer 1 Report
Through the follow-up of a large number of male patients with diabetes and non diabetes, and using multiple regression analysis and GEE analysis, this study concluded for the first time that there was a significant correlation between male muscle mass represented by ASMI and most bone related parameters in DM and non DM groups. However, the research process and the content of the paper still have the following deficiencies. First, as the control condition of diabetes group, how to ensure the similarity of blood glucose levels of patients in this group and the stability and similarity of blood glucose changes during follow-up. It is well known that different blood glucose levels and the duration of diabetes have different effects on skeletal muscle mass, bone microstructure and bone mineral density. The existence of this problem makes the subsequent data analysis have errors and unreliability. Second, the content of the abstract in the paper is a little messy and unclear. Third, the conclusion at the end of the article is lack of organization. Moreover, the final conclusion of this study is of little significance and lacks practical application value.
Reviewer 2 Report
1. Introduction: What did you hypothesize you would learn from looking at the correlation between muscle and bone parameters in diabetic and non-diabetic men? From the introduction it is unclear why you undertook to study this relationship. What would it show you about either physiology or pathophysiology of bone and muscle ?
2. Methods: Statistical analysis: Did you apply a Bonferroni correction for multiple comparisons? If not, how did you correct for multiple comparisons?
3. Methods: Since you were studying muscle and bone and diabetes, why did you not look at undercarboxylated osteocalcin, which is purported to play a major role in glucose metabolism, especially in skeletal muscle?
4. Discussion: As in the Introduction, you did not mention what you learned about bone and muscle in men with or without diabetes by looking at the bone-muscle correlation. The answers are critical to making this study clinically and physiologically meaningful
Reviewer 3 Report
Please see the Detailed information for Authors.
The authors recruit the participants from hospital that the potential for selection bias is a particular problem and the Berkesonian bias could be occur.
The definition of DM is not clear. The authors excluded patients who had taken drugs for diabetes, is it mean these patients mild symptoms of DM? How long is the duration of DM? These may affect the BMD.
Round 2
Reviewer 2 Report
The responses are generally adequate. I would recommend two further actions.First I would change the title to stress that the bone-muscle relationship appears grossly intact in diabetic men. Second, I would check with your statistician to be sure you do not need to correct for multiple comparisons within your two major groups
